# Artificial intelligence in systematic reviews: promising when appropriately used

Sanne H B van Dijk ,[1,2] Marjolein G J Brusse-Keizer,[1,3] Charlotte C Bucsán,[2,4] Job van der Palen ,[3,4] Carine J M Doggen,[1,5] Anke Lenferink [1,2,5]

¹Health Technology & Services Research, Technical Medical Centre, University of Twente, Enschede, The Netherlands
²Pulmonary Medicine, Medisch Spectrum Twente, Enschede, The Netherlands
³Medical School Twente, Medisch Spectrum Twente, Enschede, The Netherlands
⁴Cognition, Data & Education, Faculty of Behavioural, Management & Social Sciences, University of Twente, Enschede, The Netherlands
⁵Clinical Research Centre, Rijnstate Hospital, Arnhem, The Netherlands

**Correspondence to**
Dr Anke Lenferink;
a.lenferink@utwente.nl

## ABSTRACT

**Background** Systematic reviews provide a structured overview of the available evidence in medical-scientific research. However, due to the increasing medical-scientific research output, it is a time-consuming task to conduct systematic reviews. To accelerate this process, artificial intelligence (AI) can be used in the review process. In this communication paper, we suggest how to conduct a transparent and reliable systematic review using the AI tool 'ASReview' in the title and abstract screening.

**Methods** Use of the AI tool consisted of several steps. First, the tool required training of its algorithm with several prelabelled articles prior to screening. Next, using a researcher-in-the-loop algorithm, the AI tool proposed the article with the highest probability of being relevant. The reviewer then decided on relevancy of each article proposed. This process was continued until the stopping criterion was reached. All articles labelled relevant by the reviewer were screened on full text.

**Results** Considerations to ensure methodological quality when using AI in systematic reviews included: the choice of whether to use AI, the need of both deduplication and checking for inter-reviewer agreement, how to choose a stopping criterion and the quality of reporting. Using the tool in our review resulted in much time saved: only 23% of the articles were assessed by the reviewer.

**Conclusion** The AI tool is a promising innovation for the current systematic reviewing practice, as long as it is appropriately used and methodological quality can be assured.

**PROSPERO registration number** CRD42022283952.

## STRENGTHS AND LIMITATIONS OF THIS STUDY

⇒ Potential pitfalls regarding the use of artificial intelligence in systematic reviewing were identified.
⇒ Remedies for each pitfall were provided to ensure methodological quality. A time-efficient approach is suggested on how to conduct a transparent and reliable systematic review using an artificial intelligence tool.
⇒ The artificial intelligence tool described in the paper was not evaluated for its accuracy.

reviews facilitate up-to-date and accessible summaries of evidence, as they synthesise previously published results in a transparent and reproducible manner.[5 6] Hence, conclusions can be drawn that provide the highest considered level of evidence in medical research.[5 7] Therefore, systematic reviews are not only crucial in science, but they have a large impact on clinical practice and policy-making as well.[6] They are, however, highly labour-intensive to conduct due to the necessity of screening a large amount of articles, which results in a high consumption of research resources. Thus, efficient and innovative reviewing methods are desired.[8]

An open-source artificial intelligence (AI) tool 'ASReview'[9] was published in 2021 to facilitate the title and abstract screening process in systematic reviews. Applying this tool facilitates researchers to conduct more efficient systematic reviews: simulations already showed its time-saving potential.[9–11] We used the tool in the study selection of our own systematic review and came across scenarios that needed consideration to prevent loss of methodological quality. In this communication paper, we provide a reliable and transparent AI-supported systematic reviewing approach.

## BACKGROUND

Medical-scientific research output has grown exponentially since the very first medical papers were published.[1–3] The output in the field of clinical medicine increased and keeps doing so.[4] To illustrate, a quick PubMed search for 'cardiology' shows a fivefold increase in annual publications from 10 420 (2007) to 52 537 (2021). Although the medical-scientific output growth rate is not higher when compared with other scientific fields,[1–3] this field creates the largest output.[3] Staying updated by reading all published articles is therefore not feasible. However, systematic

## METHODS

We first describe how the AI tool was used in a systematic review conducted by our research group. For more detailed information regarding searches and eligibility criteria of the review, we refer to the protocol (PROSPERO registry: CRD42022283952). Subsequently, when deciding on the AI screening-related methodology, we applied appropriate remedies against foreseen scenarios and their pitfalls to maintain a reliable and transparent approach. These potential scenarios, pitfalls and remedies will be discussed in the Results section.

In our systematic review, the AI tool 'ASReview' (V.0.17.1)[9] was used for the screening of titles and abstracts by the first reviewer (SHBvD). The tool uses an active researcher-in-the-loop machine learning algorithm to rank the articles from high to low probability of eligibility for inclusion by text mining. The AI tool offers several classifier models by which the relevancy of the included articles can be determined.[9] In a simulation study using six large systematic review datasets on various topics, a Naïve Bayes (NB) and a term frequency-inverse document frequency (TF-IDF) outperformed other model settings.[10] The NB classifier estimates the probability of an article being relevant, based on TF-IDF measurements. TF-IDF measures the originality of a certain word within the article relative to the total number of articles the word appears in.[12] This combination of NB and TF-IDF was chosen for our systematic review.

Before the AI tool can be used for the screening of relevant articles, its algorithm needs training with at least one relevant and one irrelevant article (ie, prior knowledge). It is assumed that the more prior knowledge, the better the algorithm is trained at the start of the screening process, and the faster it will identify relevant articles.[9] In our review, the prior knowledge consisted of three relevant articles[13–15] selected from a systematic review on the topic[16] and three randomly picked irrelevant articles.

After training with the prior knowledge, the AI tool made a first ranking of all unlabelled articles (ie, articles not yet decided on eligibility) from highest to lowest probability of being relevant. The first reviewer read the title and abstract of the number one ranked article and made a decision ('relevant' or 'irrelevant') following the eligibility criteria. Next, the AI tool took into account this additional knowledge and made a new ranking. Again, the next top ranked article was proposed to the reviewer, who made a decision regarding eligibility. This process of AI making rankings and the reviewer making decisions, which is also called 'researcher-in-the-loop', was repeated until the predefined data-driven stopping criterion of – in our case – 100 subsequent irrelevant articles was reached. After the reviewer rejected what the AI tool puts forward as 'most probably relevant' a hundred times, it was assumed that there were no relevant articles left in the unseen part of the dataset.

The articles that were labelled relevant during the title and abstract screening were each screened on full text independently by two reviewers (SHBvD and MGJB-K, AL, JvdP, CJMD, CCB) to minimise the influence of subjectivity on inclusion. Disagreements regarding inclusion were solved by a third independent reviewer.

## RESULTS

### How to maintain reliability and transparency when using AI in title and abstract screening

A summary of the potential scenarios, and their pitfalls and remedies, when using the AI tool in a systematic review is given in table 1. These potential scenarios should not be ignored, but acted on to maintain reliability and transparency. Figure 1 shows when and where to act on during the screening process reflected by the Preferred Reporting Items for Systematic Reviews and Meta-Analyses (PRISMA) flowchart,[17] from literature search results to publishing the review.

In our systematic review, by means of broad literature searches in several scientific databases, a first set of potentially relevant articles was identified, yielding 8456 articles, enough to expect the AI tool to be efficient in the title and abstract screening (scenario ① was avoided, see table 1). Subsequently, this complete set of articles was uploaded in reference manager EndNote X9[18] and review manager Covidence,[19] where 3761 duplicate articles were removed. Given that EndNote has quite low sensitivity in identifying duplicates, additional deduplication in Covidence was considered beneficial.[20] Deduplication is usually applied in systematic reviewing,[20] but is increasingly important prior to the use of AI. Since multiple decisions regarding a duplicate article weigh more than one, this will disproportionately influence classification and possibly the results (table 1, scenario ②). In our review, a deduplicated set of articles was uploaded in the AI tool. Prior to the actual AI-supported title and abstract screening, the reviewers (SHBvD and AL, MGJB-K) trained themselves with a small selection of 74 articles. The first reviewer became familiar with the ASReview software, and all three reviewers learnt how to apply the eligibility criteria, to minimise personal influence on the article selection (table 1, scenario ③).

Defining the stopping criterion used in the screening process is left to the reviewer.[9] An optimal stopping criterion in active learning is considered a perfectly balanced trade-off between a certain cost (in terms of time spent) of screening one more article versus the predictive performance (in terms of identifying a new relevant article) that could be increased by adding one more decision.[21] The optimal stopping criterion in systematic reviewing would be the moment that screening additional articles will not result in more relevant articles being identified.[22] Therefore, in our review, we predetermined a data-driven stopping criterion for the title and abstract screening as '100 consecutive irrelevant articles' in order to prevent the screening from being stopped before or a long time after all relevant articles were identified (table 1, scenario ④).

Due to the fact that the stopping criterion was reached after 1063 of the 4695 articles, only a part of the total

van Dijk SHB, *et al. BMJ Open* 2023;**13**:e072254. doi:10.1136/bmjopen-2023-072254

**Table 1** Per-scenario overview of potential pitfalls and how to prevent these when using ASReview in a systematic review

| Potential scenario | | Pitfall | Remedy |
|---|---|---|---|
| ① | Only a small (ie, manually feasible*) number of articles (with possibly a high proportion relevant) available for screening | Time wasted by considering AI-related choices, software training and no time saved by using AI | Do not use AI: conduct manual screening |
| ② | Presence of duplicate articles in ASReview | Unequal weighing of labelled articles in AI-supported screening | Apply deduplication methods before using AI |
| ③ | Reviewer's own opinion, expertise or mistakes influence(s) AI algorithm on article selection | Not all relevant articles are included, potentially introducing selection bias | Reviewer training in title and abstract screening Perform (partial) double screening and check inter-reviewer agreement |
| ④ | AI-supported screening is stopped before or a long time after all relevant articles are found | Not all relevant articles are included, potentially introducing selection bias, or time is wasted | Formulate a data-driven stopping criterion (ie, number of consecutive irrelevant articles) |
| ⑤ | AI-related choices not (completely) described | Irreproducible results, leading to a low-quality systematic review | Describe and substantiate the choices that are made |
| ⑥ | Study selection is not transparent | Irreproducible results (black box algorithm), leading to a low-quality systematic review | Publish open data (ie, extracted file with all decisions) |

*What is considered manually feasible is highly context-dependent (ie, the intended workload and/or reviewers available).

number of articles was seen. Therefore, this approach might be sensitive to possible mistakes when articles are screened by only one reviewer, influencing the algorithm, possibly resulting in an incomplete selection of articles (table 1, scenario ③).[23] As a remedy, second reviewers (AL, MGJB-K) checked 20% of the titles and abstracts

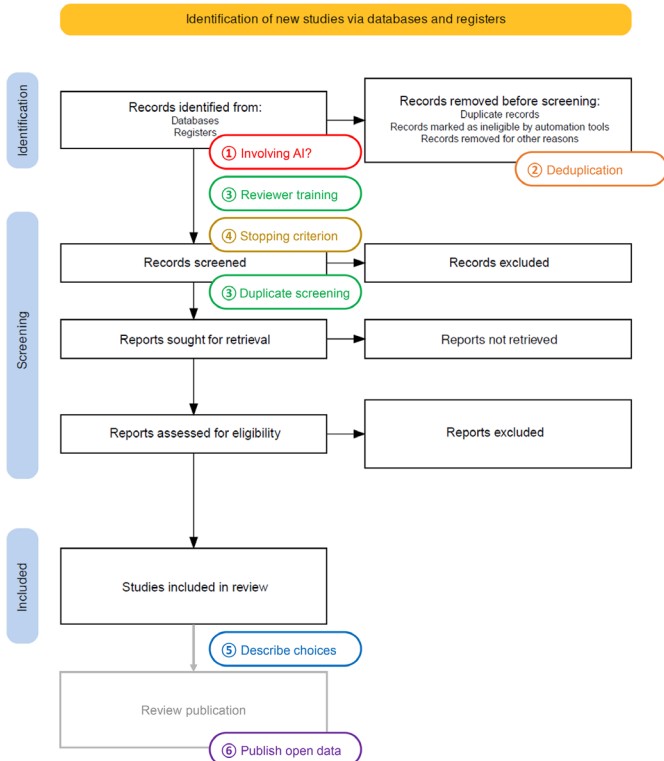

**Figure 1** Flowchart showing when and where to act on when using ASReview in systematic reviewing. Adapted the PRISMA flowchart from Haddaway *et al*.[17]

seen by the first reviewer. This 20% had a comparable ratio regarding relevant versus irrelevant articles over all articles seen. The percentual agreement and Cohen's Kappa (κ), a measure for the inter-reviewer agreement above chance, were calculated to express the reliability of the decisions taken.[24] The decisions were agreed in 96% and κ was 0.83. A κ equal of at least 0.6 is generally considered high,[24] and thus it was assumed that the algorithm was reliably trained by the first reviewer.

The reporting of the use of the AI tool should be transparent. If the choices made regarding the use of the AI tool are not entirely reported (table 1, scenario ⑤), the reader will not be able to properly assess the methodology of the review, and review results may even be graded as low-quality due to the lack of transparent reporting. The ASReview tool offers the possibility to extract a data file providing insight into all decisions made during the screening process, in contrast to various other 'black box' AI-reviewing tools.[9] This file will be published alongside our systematic review to provide full transparency of our AI-supported screening. This way, the screening with AI is reproducible (remedy to scenario ⑥, table 1).

### Results of AI-supported study selection in a systematic review

We experienced an efficient process of title and abstract screening in our systematic review. Whereas the screening was performed with a database of 4695 articles, the stopping criterion was reached after 1063 articles, so 23% were seen. Figure 2A shows the proportion of articles identified as being relevant at any point during the AI-supported screening process. It can be observed that the articles are indeed prioritised by the active learning algorithm: in the beginning, relatively many relevant articles were found, but this decreased as the stopping criterion (vertical red

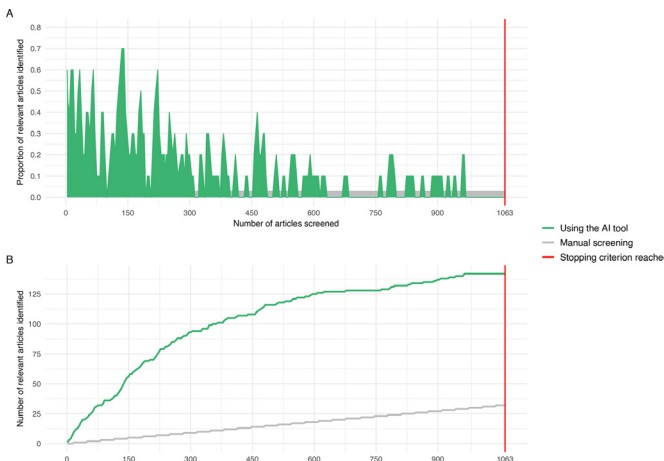

**Figure 2** Relevant articles identified after a certain number of titles and abstracts were screened using the AI tool compared with manual screening.

line) was approached. Figure 2B compares the screening progress when using the AI tool versus manual screening. The moment the stopping criterion was reached, approximately 32 records would have been found when the titles and abstract would have been screened manually, compared with 142 articles labelled relevant using the AI tool. After the inter-reviewer agreement check, 142 articles proceeded to the full text reviewing phase, of which 65 were excluded because these were no articles with an original research format, and three because the full text could not be retrieved. After full text reviewing of the remaining 74 articles, 18 articles from 13 individual studies were included in our review. After snowballing, one additional article from a study already included was added.

## DISCUSSION

In our systematic review, the AI tool considerably reduced the number of articles in the screening process. Since the AI tool is offered open source, many researchers may benefit from its time-saving potential in selecting articles. Choices in several scenarios regarding the use of AI, however, are still left open to the researcher, and need consideration to prevent pitfalls. These include the choice whether or not to use AI by weighing the costs versus the benefits, the importance of deduplication, double screening to check inter-reviewer agreement, a data-driven stopping criterion to optimally use the algorithm's predictive performance and quality of reporting of the AI-related methodology chosen. This communication paper is, to our knowledge, the first elaborately explaining and discussing these choices regarding the application of this AI tool in an example systematic review.

The main advantage of using the AI tool is the amount of time saved. Indeed, in our study, only 23% of the total number of articles were screened before the predefined stopping criterion was met. Assuming that all relevant articles were found, the AI tool saved 77% of the time

for title and abstract screening. However, time should be invested to become acquainted with the tool. Whether the expected screening time saved outweighs this time investment is context-dependent (eg, researcher's digital skills, systematic reviewing skills, topic knowledge). An additional advantage is that research questions previously unanswerable due to the insurmountable number of articles to screen in a 'classic' (ie, manual) review, now actually are possible to answer. An example of the latter is a review screening over 60 000 articles,[25] which would probably never have been performed without AI supporting the article selection.

Since the introduction of the ASReview tool in 2021, it was applied in seven published reviews.[25–31] An important note to make is that only one[25] clearly reported AI-related choices in the methods and a complete and transparent flowchart reflecting the study selection process in the Results section. Two reviews reported a relatively small number (<400) of articles to screen,[26 27] of which more than 75% of the articles were screened before the stopping criterion was met, so the amount of time saved was limited. Also, three reviews reported many initial articles (>6000)[25 28 29] and one reported 892 articles,[31] of which only 5%–10% needed to be screened. So in these reviews, the AI tool saved an impressive amount of screening time. In our systematic review, 3% of the articles were labelled relevant during the title and abstract screening and eventually, <1% of all initial articles were included. These percentages are low, and are in line with the three above-mentioned reviews (1%–2% and 0%–1%, respectively).[25 28 29] Still, relevancy and inclusion rates are much lower when compared with 'classic' systematic reviews. A study evaluating the screening process in 25 'classic' systematic reviews showed that approximately 18% was labelled relevant and 5% was actually included in the reviews.[32] This difference is probably due to more narrow literature searches in 'classic' reviews for feasibility purposes compared with AI-supported reviews, resulting in a higher proportion of included articles.

In this paper, we show how we applied the AI tool, but we did not evaluate it in terms of accuracy. This means that we have to deal with a certain degree of uncertainty. Despite the data-driven stopping criterion there is a chance that relevant articles were missed, as 77% was automatically excluded. Considering this might have been the case, first, this could be due to wrong decisions of the reviewer that would have undesirably influenced the training of the algorithm by which the articles were labelled as (ir)relevant and the order in which they were presented to the reviewer. Relevant articles could have therefore remained unseen if the stopping criterion was reached before they were presented to the reviewer. As a remedy, in our own systematic review, of the 20% of the articles screened by the first reviewer, relevancy was also assessed by another reviewer to assess inter-reviewer reliability, which was high. It should be noted, though, that 'classic' title and abstract screening is not necessarily better than using AI, as medical-scientific researchers tend

to assess one out of nine abstracts wrongly.[32] Second, the AI tool may not have properly ranked highly relevant to irrelevant articles. However, given that simulations proved this AI tool's accuracy before[9–11] this was not considered plausible. Since our study applied, but did not evaluate, the AI tool, we encourage future studies evaluating the performance of the tool across different scientific disciplines and contexts, since research suggests that the tool's performance depends on the context, for example, the complexity of the research question.[33] This could not only enrich the knowledge about the AI tool, but also increases certainty about using it. Also, future studies should investigate the effects of choices made regarding the amount of prior knowledge that is provided to the tool, the number of articles defining the stopping criterion, and how duplicate screening is best performed, to guide future users of the tool.

Although various researcher-in-the-loop AI tools for title and abstract screening have been developed over the years,[9 23 34] they often do not develop into usable mature software,[34] which impedes AI to be permanently implemented in research practice. For medical-scientific research practice, it would therefore be helpful if large systematic review institutions, like Cochrane and PRISMA, would consider to 'officially' make AI part of systematic reviewing practice. When guidelines on the use of AI in systematic reviews are made available and widely recognised, AI-supported systematic reviews can be uniformly conducted and transparently reported. Only then we can really benefit from AI's time-saving potential and reduce our research time waste.

## CONCLUSION

Our experience with the AI tool during the title and abstract screening was positive as it has highly accelerated the literature selection process. However, users should consider applying appropriate remedies to scenarios that may form a threat to the methodological quality of the review. We provided an overview of these scenarios, their pitfalls and remedies. These encourage reliable use and transparent reporting of AI in systematic reviewing. To ensure the continuation of conducting systematic reviews in the future, and given their importance for medical guidelines and practice, we consider this tool as an important addition to the review process.

**Contributors** SHBvD proposed the methodology and conducted the study selection. MGJB-K, CJMD and AL critically reflected on the methodology. MGJB-K and AL contributed substantially to the study selection. CCB, JvdP and CJMD contributed to the study selection. The manuscript was primarily prepared by SHBvD and critically revised by all authors. All authors read and approved the final manuscript.

**Funding** The systematic review is conducted as part of the RE-SAMPLE project. RE-SAMPLE has received funding from the European Union's Horizon 2020 research and innovation programme (grant agreement no. 965315).

**Competing interests** None declared.

**Ethics approval** Not applicable.

**Provenance and peer review** Not commissioned; externally peer reviewed.

**ORCID iDs**
Sanne H B van Dijk http://orcid.org/0000-0003-1727-0608
Job van der Palen http://orcid.org/0000-0003-1071-6769
Anke Lenferink http://orcid.org/0000-0002-2276-5691

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
