## [Reviewer comments · BMJ Open]

ARTICLE DETAILS

TITLE (PROVISIONAL)	Artificial intelligence in systematic reviews: promising when appropriately used
AUTHORS	van Dijk, Sanne H B; Brusse-Keizer, Marjolein; Bucsán, Charlotte; van der Palen, Job; Doggen, Carine J.M.; Lenferink, Anke

VERSION 1 – REVIEW

REVIEWER	Aceves-Martins, Magaly University of Aberdeen, Health Services Research Unit
REVIEW RETURNED	05-May-2023

GENERAL COMMENTS	I really enjoyed reading this communication article entitled “A modern era systematic review using artificial intelligence: considerations to ensure methodological quality”, which aims to suggest how to conduct a transparent and reliable systematic review using the AI tool ‘ASReview’ in the title and abstract screening. The authors performed a fantastic job highlighting some of the challenges/opportunities of integrating new AI tools (such as ASReview) into the systematic review process. However, some aspects of the communication article could be strengthened and clarified to make it worthwhile for future guidance on using such tools. I hope my comments/suggestions/doubts help authors address some of these inconsistencies and improve their manuscript. Title: Please revise the title as readability could be improved. ABSTRACT: As a clarification, AI can be used in several steps of the systematic reviewing process, not only in title and abstract screening. I suggest rephrasing the following sentence to clarify this. “To accelerate this process, artificial intelligence (AI) can be used in the screening of titles and abstracts”. BACKGROUND: Page 7, line 31: SRs are not only labour-intensive but also resource-consuming. This could also be clarified. METHODS: Page 8, lines 52-57: Is there any rationale for selecting three relevant and three irrelevant studies? A previous sentence in this paragraph stated, “It is assumed that the more prior knowledge, the better the algorithm is trained”. However, a debate in the area at the moment is questioning if more studies are better, while better for training the algorithm. Page 9, lines 11-17: This paragraph might need to be clarified for readers unfamiliar with ASReview, as it would seem that the reviewer makes different rankings after the AI makes the first ranking. Adding the clarification that ASReview will do the ranking as it uses a machine-learning approach using a human in the loop.
---

	It would be important that, at this point, it is also clarified if the reviewer followed a data-driven strategy to train the algorithm (i.e., the reviewer decided to stop after the algorithm retrieves an x amount of consecutive irrelevant papers) and to provide a rationale for this. Page 9, lines 17-24: In this second classification, how did the reviewers select 100 subsequent irrelevant articles as a threshold? Page 9, lines 27-34: What about those abstracts labelled as irrelevant? Was there any cross-checking in these? Would it be beneficial (or not) to cross-check these? RESULTS: Figure 1: I found Figure 1 confusing. It would be clearer if the two flowcharts (the boxes with text and the colour bubbles) could be integrated within the same figure. Also, after AI-support screening, the following step is to review the publication, which jumps up to review publication, missing several stages of the systematic review process. Creating a more comprehensive (including both flowcharts in one and including all the review processes) figure would benefit this work. As authors are using ASReview, which is only aiding at the screening titles and abstracts stage, they must specify this in the results. Rather than "How to maintain reliability and transparency when using AI in systematic reviewing", it should be clear that is reliability and transparency when using AI to screen titles and abstracts in systematic reviews. Table 1 needs to be put in the right place. While reading Table 1, several queries were raised (answered in the text), so it might be helpful to place the table after the text explaining it. I understand that authors are commenting on their experiences while using this tool. However, some potential scenarios are linked more to the systematic review methods rather than the usage of AI. For example, Potential scenario #2, "Presence of duplicate articles in ASReview". Deduplication is suggested by Cochrane and PRISMA, even if not using AI tools. So, the remedy applies to AI but also to human reviewing. When training the algorithm, I can see how important this is, but more information should be provided for the reader at this point. So that is why it would be good to have the text before the table. Potential scenario 1: What is considered "manually feasible"? This also depends on the available resources (e.g., the number of reviewers assigned), so it might differ from case to case. Page 11, lines 4 to 6: Why were two reference managers (endnote and Covidence) where used? DISCUSSION/CONCLUSIONS: Although AI saved time in the title and abstract screening, how long were reviewers trained for? Practically, this has been one of the main barriers for some systematic review teams to include AI in their regular practice. The time/cost of training their staff might be an opportunity window but is also a barrier. I would appreciate some discussion on this matter.
--	--

REVIEWER	Muthu, Sathish Orthopaedic Research Group
REVIEW RETURNED	07-May-2023

GENERAL COMMENTS	Dear Authors Congratulations on the communication paper to address the issues with AI-powered screening tools (ASReview) in maintaining
--

	the methodological quality of the systematic reviews adopting them. 1. I see your concerns that would arise in incorporating such tools and the methods to maintain the methodological quality in the same are well discussed. I see that you have missed another article by Muthu et al. 2022 (10.1007/s00264-022-05672-y) that discussed the issues with the efficiency of the AI-assisted platform, in fact, the ASReview platform itself. In his article, he utilized the platform for three scenarios based on the difficulty of the research question and demonstrated the efficiency of the system varies with the difficulty in the context of the research question being employed. It is imperative that this concern needs to be discussed in this paper and methods to tackle this situation also need elaboration. It is noted that the number of articles to make the hard stop to fetch the relevant article depends both on the training of the AI algorithm and the difficulty put forth to the system by the context of the research question. 2. The figure needs to be redrawn to make it more meaningful and visually appealing to convey the context to the readers. A line diagram demonstrating the depletion of the relevant article from the selected list with the progression of algorithm training might be utilized. 3. The process of screening is duplicated to avoid selection bias and it has been done so far till now for the entire list of articles that needs screening. In this context, from the given solution for the selection bias, it has been suggested that only 20% of the articles are reviewed by the second reviewer and assessed with Cohen's kappa which seems so less. Given the reviewer has a 0.61 kappa value, it may be considered high but still, the training of the algorithm with the given kappa value might be different and the ranking of the articles and its selection might be varied. Hence I suggest two reviewers use the tool and make their individual selections and then compare the list of selected articles for the discrepancy and resolve the issue with third reviewer and also report the kappa value between the two to know the level of symmetry in the algorithm training along with the number of articles screening in total by the individual reviewers to achieve the predefined hard stop value of irrelevant articles screen (100 in the given example).
--	---

VERSION 1 – AUTHOR RESPONSE

Reviewer 1: TITLE: 1. Please revise the title as readability could be improved. Response: We revised the title to: 'Artificial intelligence in systematic reviews: promising when appropriately used' (page 1, line 3). ABSTRACT: 2. As a clarification, AI can be used in several steps of the systematic reviewing process, not only in title and abstract screening. I suggest rephrasing the following sentence to clarify this. "To accelerate this process, artificial intelligence (AI) can be used in the screening of titles and abstracts". Response: We rephrased this sentence into 'In this communication paper, we suggest how to conduct a transparent and reliable systematic review using the AI tool 'ASReview' in the title and abstract screening' (page 2, line 5-7). BACKGROUND: 3. Page 7, line 31: SRs are not only labour-intensive but also resource-consuming. This could also be clarified. Response: We wanted to point out here that systematic reviews are resource-consuming because they are highly labour-intensive. Therefore, we added this information to this sentence 'They are, however, highly labour-intensive to conduct due to the necessity of screening a large amount of articles, which results in a high consumption of research resources' (page 3, line 12-14). METHODS: 4. Page 8, lines 52-57: Is there

any rationale for selecting three relevant and three irrelevant studies? A previous sentence in this paragraph stated, "It is assumed that the more prior knowledge, the better the algorithm is trained". However, a debate in the area at the moment is questioning if more studies are better, while better for training the algorithm. Response: There was no clear guidance on selecting the number of prior knowledge articles when using ASReview at that moment. However, our rationale to select three relevant and three irrelevant articles to feed the prior knowledge of ASReview was mainly to optimise the efficiency. For us, this meant: selecting enough prior knowledge to get the screening process going (not to wait too long before arriving at the first relevant article), but not investing too much time upfront to select the prior knowledge. In our opinion, future studies must define what choices are considered efficient and appropriate. Following the reviewer's comment, we emphasised this in the discussion section where we discuss topics for future research: 'Also, future studies should investigate the effects of choices made, regarding both the amount of prior knowledge that is provided to the tool and the number of articles defining the stopping criterion, to guide future users of the tool.' (page 12, line 5-8).

5. Page 9, lines 11-17: This paragraph might need to be clarified for readers unfamiliar with ASReview, as it would seem that the reviewer makes different rankings after the AI makes the first ranking. Adding the clarification that ASReview will do the ranking as it uses a machine-learning approach using a human in the loop. It would be important that, at this point, it is also clarified if the reviewer followed a data-driven strategy to train the algorithm (i.e., the reviewer decided to stop after the algorithm retrieves an x amount of consecutive irrelevant papers) and to provide a rationale for this. Response: We think we managed to clarify this paragraph for readers who are unfamiliar with ASReview, based on the reviewer's comment by simplifying some sentences and mentioning the term that was suggested by the reviewer: 'researcher-in-the-loop': 'The first reviewer read the title and abstract of the number one ranked article and made a decision ('relevant' or 'irrelevant') following the eligibility criteria. Next, the AI tool took into account this additional knowledge and made a new ranking. Again, the next top ranked article was proposed to the reviewer, who made a decision regarding eligibility. This process of AI making rankings and the reviewer making decisions, which is also called 'researcher-in-the-loop', was repeated until the predefined data-driven stopping criterion of – in our case - 100 subsequent irrelevant articles was reached. After the reviewer rejected what the AI tool puts forward as 'most probably relevant' a hundred times, it was assumed that there were no relevant articles left in the unseen part of the dataset.' (page 5, line 3-9). Furthermore, the point regarding the stopping criterion is now addressed by adding some additional explanation for the reader unfamiliar with ASReview regarding the rationale behind the stopping criterion: 'After the reviewer rejected what the AI tool puts forward as 'most probably relevant' a hundred times, it was assumed that there were no relevant articles left in the unseen part of the dataset.' (page 5, line 9-11).

6. Page 9, lines 17-24: In this second classification, how did the reviewers select 100 subsequent irrelevant articles as a threshold? Response: Unfortunately, there is no clear guidance on a threshold for the number of irrelevant articles (please also see our response to question 4). The decision to select 100 subsequent articles as a threshold was based on balancing 1) the chance of stopping too early, resulting in relevant articles missed; and 2) continue screening too long and losing efficiency. We emphasized the important role of future studies in defining an 'ideal' threshold in the discussion: 'Also, future studies should investigate the effects of choices made regarding the amount of prior knowledge that is provided to the tool, the number of articles defining the stopping criterion, and how duplicate screening is best performed, to guide future users of the tool.' (page 12, line 8-11).

7. Page 9, lines 27-34: What about those abstracts labelled as irrelevant? Was there any crosschecking in these? Would it be beneficial (or not) to cross-check these? Response: The scope of the current article is how to conduct a transparent systematic review using the AI tool and not to test its actual performance. That is why we did not cross-check all abstracts labelled as irrelevant. However, from the abstracts labelled irrelevant by the first reviewer, 20% was checked by a second reviewer, but this was performed to minimise the influence of subjectivity on inclusion (see also page...). Next to this in the discussion section we did emphasise the need of future studies evaluating the AI tools performance and refer to the simulation studies conducted, which concluded high accuracy for the combination of the Naïve Bayes classifier and TF-IDF (that is, after screening 10% of the abstracts,

between 65-100% of all relevant articles were found) (page 12, line 3 (reference 9-11)). RESULTS: 8. Figure 1: I found Figure 1 confusing. It would be clearer if the two flowcharts (the boxes with text and the colour bubbles) could be integrated within the same figure. Also, after AI-support screening, the following step is to review the publication, which jumps up to review publication, missing several stages of the systematic review process. Creating a more comprehensive (including both flowcharts in one and including all the review processes) figure would benefit this work. Response: We understand this remark and agree with the reviewer that some review steps were skipped in this figure. Therefore, we have now decided to take an empty PRISMA flow diagram (10.1002/cl2.1230) reflecting the complete systematic review process and place the scenarios described in our paper at the correct place in this review process. We think that the new figure (Figure 1) better reflects the role of ASReview in the complete review process compared with the previous version, so we thank the reviewer for the comment. 9. As authors are using ASReview, which is only aiding at the screening titles and abstracts stage, they must specify this in the results. Rather than "How to maintain reliability and transparency when using AI in systematic reviewing ", it should be clear that is reliability and transparency when using AI to screen titles and abstracts in systematic reviews. Response: We changed the first subheading in the results from 'How to maintain reliability and transparency when using AI in systematic reviewing' to 'How to maintain reliability and transparency when using AI in title and abstract screening' 10. Table 1 needs to be put in the right place. While reading Table 1, several queries were raised (answered in the text), so it might be helpful to place the table after the text explaining it. I understand that authors are commenting on their experiences while using this tool. However, some potential scenarios are linked more to the systematic review methods rather than the usage of AI. For example, Potential scenario #2, "Presence of duplicate articles in ASReview". Deduplication is suggested by Cochrane and PRISMA, even if not using AI tools. So, the remedy applies to AI but also to human reviewing. When training the algorithm, I can see how important this is, but more information should be provided for the reader at this point. So that is why it would be good to have the text before the table. Response: We agree with the reviewer's comment. Since readers are best provided with essential information first, we have now placed Table 1 directly after the text explaining the table (page 6-8). 11. Potential scenario 1: What is considered "manually feasible"? This also depends on the available resources (e.g., the number of reviewers assigned), so it might differ from case to case. Response: We agree. Because this feasibility is very context-specific, we did not further define what is considered "manually feasible". However, we decided to add a footnote to Table 1, stressing that whether or not it is manually feasible, strongly depends on this context: 'What is considered manually feasible is highly context-dependent (i.e., the intended workload and/or number reviewers available)' (page 8, line 6- 7). 12. Page 11, lines 4 to 6: Why were two reference managers (endnote and Covidence) used? Response: EndNote was used to first combine the search results from the different databases and second to deduplicate. Because EndNote has quite a low sensitivity in identifying duplicates, also Covidence was used. Covidence has a high sensitivity in identifying duplicates (10.1186/s13643-021-01583-y). We have added a sentence explaining why both reference managers were used for deduplication: 'Given that EndNote has quite low sensitivity in identifying duplicates, additional deduplication in Covidence was considered beneficial' (page 6, line 12-14). DISCUSSION/CONCLUSIONS: 13. Although AI saved time in the title and abstract screening, how long were reviewers trained for? Practically, this has been one of the main barriers for some systematic review teams to include AI in their regular practice. The time/cost of training their staff might be an opportunity window but is also a barrier. I would appreciate some discussion on this matter. Response: We thank the reviewer for this important remark. We think that becoming acquainted with new software can indeed be a barrier, but how much of a barrier this is, is context-dependent. This depends on e.g., digital skills of the reviewers, reviewers' background knowledge with regard to systematic reviewing, and considering the costs (in this case, training time) versus the benefit (improved efficiency). Also, for one's second ASReview-supported review, additional training may not be needed anymore. It is up to the reviewers themselves to determine whether they think it is worthwhile to invest time. We added this point to pitfall 1 (Table 1) and clarified that the choice of using AI here is a question of costs versus benefits (page 10, line 6-9). Later in the

discussion section, we now shed some additional light on this matter by adding the following sentences: 'However, time should be invested to become acquainted with the tool. Whether the expected screening time saved outweighs this time investment is context-dependent (e.g., researcher's digital skills, systematic reviewing skills, topic knowledge)' (page 10, line 15-18).

Reviewer 2: 1. I see your concerns that would arise in incorporating such tools and the methods to maintain the methodological quality in the same are well discussed. I see that you have missed another article by Muthu et al. 2022 (10.1007/s00264-022-05672-y) that discussed the issues with the efficiency of the AI-assisted platform, in fact, the ASReview platform itself. In his article, he utilized the platform for three scenarios based on the difficulty of the research question and demonstrated the efficiency of the system varies with the difficulty in the context of the research question being employed. It is imperative that this concern needs to be discussed in this paper and methods to tackle this situation also need elaboration. It is noted that the number of articles to make the hard stop to fetch the relevant article depends both on the training of the AI algorithm and the difficulty put forth to the system by the context of the research question. Response: We thank the reviewer for this suggestion and read the publication by Muthu with much interest. We added the information in the paragraph discussing options for future research, as we think the point made in the publication the reviewer suggested is indeed relevant to include in our manuscript: 'Since our study applied, but did not evaluate, the AI tool, we encourage future studies evaluating the performance of the tool across different scientific disciplines and contexts, since research suggests that the tool's performance depends on the context, for example, the complexity of the research question' (page 12, line 4-7).

2. The figure needs to be redrawn to make it more meaningful and visually appealing to convey the context to the readers. A line diagram demonstrating the depletion of the relevant article from the selected list with the progression of algorithm training might be utilized. Response: We agree that a line plot would make the figure more clear. We adjusted the figure and added the line plot (B) below the area plot (A), so that the reader gains insight in the review process this way. We also added some text to explain the renewed figure: 'Figure 2A shows the proportion of articles identified as being relevant at any point during the AI-supported screening process. It can be observed that the articles are indeed prioritised by the active learning algorithm: in the beginning, relatively many relevant articles were found, but this decreased as the stopping criterion (vertical red line) was approached. Figure 2B compares the screening progress when using the AI tool versus manual screening. The moment the stopping criterion was reached, approximately 32 records would have been found when the titles and abstract would have been screened manually, compared to 142 articles labelled relevant using the AI tool.' (page 8 line 12- - page 9, line 6).

3. The process of screening is duplicated to avoid selection bias and it has been done so far till now for the entire list of articles that needs screening. In this context, from the given solution for the selection bias, it has been suggested that only 20% of the articles are reviewed by the second reviewer and assessed with Cohen's kappa which seems so less. Given the reviewer has a 0.61 kappa value, it may be considered high but still, the training of the algorithm with the given kappa value might be different and the ranking of the articles and its selection might be varied. Hence I suggest two reviewers use the tool and make their individual selections and then compare the list of selected articles for the discrepancy and resolve the issue with third reviewer and also report the kappa value between the two to know the level of symmetry in the algorithm training along with the number of articles screening in total by the individual reviewers to achieve the predefined hard stop value of irrelevant articles screen (100 in the given example). Response: Indeed for the methodologic quality of a systematic reviews it is warranted to (partly) duplicate screening. In our review we choose to duplicate screening for 20% of the abstracts screened (see page 7, line 8-18). That is, ASReview's final ranking from the first reviewer was used, and from all articles that were screened by a person, every 9th and 10th article in that ranking was checked by a second reviewer. Due to the high kappa value of 0.83 (instead of 0.61) we decided that to increase efficiency we did not need to continue the duplicate screening. Nevertheless, we appreciate the reviewer's suggestion. Although we acknowledge the value of two reviewers both screening in ASReview until reaching the stopping criterion, this is not only a choice in the context of ASReview, but predominantly in the context of availability of resources(that is, reviewer availability). Given that our resources were limited,

we did not conduct our study the way the reviewer suggests us to do. Also, we think our approach was efficient and appropriate, given the high inter-reviewer agreement. We do, however, recognize that other choices can be made, considering the same pitfalls and remedies (Table 1). To clarify that our systematic review has been taken as an example rather than as the 'ideal' AI-supported review, we changed the wording in the discussion section at some places (page 10, line 9-11 ('example systematic review'); page 11, line 22-24 ('our own systematic review,')). By addressing the reviewers' comments, we went over the word limit of 2,500 in this new version compared to the original uploaded version. We hope the editor values the words added, as we think these contribute to a clarified and more complete manuscript. We hope that we have adequately addressed the editor's and reviewers' comments and look forward to hearing from you.

VERSION 2 – REVIEW

REVIEWER	Aceves-Martins, Magaly University of Aberdeen, Health Services Research Unit
REVIEW RETURNED	21-Jun-2023

GENERAL COMMENTS	Thanks for responding to all of the comments previously raised. I think the communication reads better now, and I have no doubt this will be useful to future researchers including AI while doing a systematic review. As a minor comment, I would suggest matching the colours scheme from the bubbles and numbers in Table 1, with those presented in the PRISMA flowchart so it is easier to associate.
---

REVIEWER	Muthu, Sathish Orthopaedic Research Group
REVIEW RETURNED	22-Jun-2023

GENERAL COMMENTS	Congratulations to the authors for addressing all the concerns raised in the previous round of review. Now I would recommend the paper for publication.
---